# Overexpression of GPX2 gene regulates the development of porcine preadipocytes and skeletal muscle cells through MAPK signaling pathway

Chunguang Zhang[1☉], Lei Wang[2☉], Lei Qin[1☉], Yunyan Luo[1], Zuochen Wen[1], Akpaca Samson Vignon[1], Chunting Zheng[1], Xueli Zhu[1], Han Chu[1], Shifan Deng[1], Liang Hong[1,3], Jianbin Zhang[1,3], Hua Yang[1], Jianbo Zhang[2], Yuhong Ma[2], Guofang Wu[2], Chao Sun[3], Xin Liu[4]*, Lei Pu[1,3]*

1 Tianjin Key Laboratory of Agricultural Animal Breeding and Healthy Husbandry, College of Animal Science and Veterinary Medicine, Tianjin Agricultural University, Tianjin, 300392, China, 2 State Key Laboratory of Plateau Ecology and Agriculture, Department of Animal Science and Veterinary Medicine, Qinghai University, Xining, China, 3 Tianjin modern Tianjiao Agricultural Technology Co, LTD, Tianjin Key Laboratory of Green Ecological Feed, Tianjin, China, 4 Institute of Animal Science, Chinese Academy of Agricultural Sciences, Beijing, China

☉ These authors contributed equally to this work.
* firstliuxin@163.com (XL); puleiwork@tjau.edu.cn (LP)

**Data Availability Statement:** All relevant data are within the paper and its Supporting Information files.

## Abstract

Glutathione peroxidase 2 (GPX2) is a selenium-dependent enzyme and protects cells against oxidative damage. Recently, GPX2 has been identified as a candidate gene for backfat and feed efficiency in pigs. However, it is unclear whether GPX2 regulates the development of porcine preadipocytes and skeletal muscle cells. In this study, adenoviral gene transfer was used to overexpress *GPX2*. Our findings suggest that overexpression of *GPX2* gene inhibited proliferation of porcine preadipocytes. And the process is accompanied by the reduction of the p-p38. *GPX2* inhibited adipogenic differentiation and promoted lipid degradation, while ERK1/2 was reduced and p-p38 was increased. Proliferation of porcine skeletal muscle cells was induced after *GPX2* overexpression, was accompanied by activation in JNK, ERK1/2, and p-p38. Overexpression methods confirmed that *GPX2* has a promoting function in myoblastic differentiation. ERK1/2 pathway was activated and p38 was suppressed during the process. This study lays a foundation for the functional study of *GPX2* and provides theoretical support for promoting subcutaneous fat reduction and muscle growth.

## Introduction

As a selenoenzyme, GPX2 is a member of the glutathione peroxidases family and is essential for protecting cellular development because it efficiently scavenges accumulated hydroperoxides and lipid peroxides [1]. GPX2 was first identified as a GPX protein that was particular to

**Funding:** This study was supported by project funding provided by Tianjin Excellent Agricultural Science and Technology Special Commissioner and Support Projects to LP [23ZYCGSN00230], National Undergraduate Innovation and Entrepreneurship Training Program to LP [202310061006], Tianjin Key Laboratory of Green ecological feed to LP [TJ202302], Anhui Natural Science Foundation Project to GX and LP [2208085MC76], Key Research, the Tianjin Pig Industry Technology System Innovation Team, China to JZ [ITTPRS2024006], Key R&D and Transformation Program of Qinghai Province Science and Technology Assistance Program for Qinghai, China to GW [2023-NK-141], National Swine Industry Technology System to XL [CARS-35], and Agricultural Science and Technology Innovation Program to XL [ASTIP-IAS02].

**Competing interests:** The authors have declared that no competing interests exist.

the gastrointestinal tract, it was known as GSHPx-GI [2]. However, the enzyme has been found in a wide range of cells and organs, including the liver, breast, and lungs [1]. *GPX2* expression was shown to be elevated in response to oxidative stress in the body or cells [3]. When *GPX2* is overexpressed, there is an increase in the ROS scavenger activity and a reduction in oxidative stress [4].

Many research have demonstrated that *GPX2* has been linked to several roles in muscle and adipose tissue [5,6]. The active center of GPX2 contains selenocysteine (SeCys). Recently, it has been discovered that proteins containing SeCys have a role in the differentiation of skeletal muscle [7,8]. A genome-wide association study (GWAS) suggested that GPX2 plays an major role in residual feed intake (RFI) trait [9]. Additionally, our earlier research shown a substantial correlation between the kind of single nucleotide polymorphism (SNP) mutation in the *GPX2* gene and the average daily gain and thickness of the backfat in Duroc pigs [10]. Nevertheless, little research has been done on the function of *GPX2* genes in the formation of skeletal muscle in pig adipocytes.

The extracellular signal-regulated kinases 1/2 (ERK1/2), the c-jun N-terminal kinase (JNK1/2/3), and p38 are components of the mitogen-activated protein kinase (MAPK) signaling pathway, which controls a variety of biological processes like cell differentiation and proliferation [11]. In this work, we investigated the impact of *GPX2* on skeletal muscle cells and porcine preadipocytes by doing a few studies and looking at the modifications to the MAPK pathway. The basis for investigating the role of antioxidant genes in muscle and subcutaneous fat was established by this work.

## Materials and methods

Our research was approved by the Ethics Committee of Tianjin Agricultural University (2022LLSC27), which conforms to the ethical requirements of animal. All the procedures were carried out under sodium pentobarbital anesthesia, and the pigs were executed while profoundly unconscious. The sedated animals were really put to death by transection of the carotid arteries, after which the longest dorsal muscle and subcutaneous adipose tissues were removed.

### Porcine preadipocytes culture and adipogenic differentiation

Piglet subcutaneous adipose tissues were removed in an aseptic manner, and PBS with a high dose of penicillin (300 U/mL) and streptomycin (300 ng/mL) was used to wash the tissues. Next, collagenase type I (Sigma, St. Louis, MO, USA) was used to chop and digest the adipose tissues. To stop the digestive process, an equivalent volume of serum culture solution was added, then filtered each material through an 80 mesh and 200 mesh sieves in turn. Centrifugation was used for 10 minutes at 2000 rpm/min to obtain the filtrate. After removing the supernatant, the culture solution was added. These cells were resuspended, and incubated in the incubator (37°C, 5% $CO_2$). And cells were treated with a lipogenic differentiation induction solution DMEM/F12 (Gibco, Grand Island, NY, USA) (containing 0.5 mM IBMX, 1 µM DEX, and 5 µg/mL insulin) to induce adipogenic differentiation for 2 days, followed by subsequent treatment.

### Porcine skeletal muscle cells culture and myogenic differentiation

The piglet's longest dorsal muscle was removed while it was still aseptic and cleaned with PBS that had a high concentration of both penicillin and streptomycin, the double antibiotics. Visible blood vessels were removed, and muscle tissue was cut into 1 $mm^3$ pieces, add type I/II collagenase (mixed 1:1), then transferred into digestion bottles. After the digestion process was

stopped, sieves of 80 and 200 mesh were used. Centrifugation was used to gather the filtrate, and it ran for 10 minutes at 2000 rpm/min. Centrifugation, incubated in 37°C and 5% $CO_2$ incubator. After inducing myoblastic differentiation in the cells for two days using the myogenic differentiation induction solution, the cells were harvested after six days.

## GPX2 adenovirus overexpressed

Construction of GPX2 adenovirus overexpression vector (pAV[Exp]-EF1A>{Sus scrofa GPX2 (ns)}:P2A:EGFP). Add DMEM/F12 medium and observes fluorescent expression. Harvest virus to achieve $>10^{10}$ IFU/mL Virus titer of concentration. GPX2 adenovirus was added into the cell passage (30 mm in diameter), while attempting to identify the proliferation impact. After the cells reached 70% density, 2 μL of GPX2 adenovirus was introduced.

Further induced differentiation occurs when cells reach a specific density. It was discovered that *GPX2* overexpression had an impact on cell differentiation.

## EdU proliferation staining

Following cell passage, the cells were grown for 24 hours in order to detect EdU growth. EdU working solution (400 μL of 50 uM) (Ribobio, Guangzhou, China) was added to the Petri dish, and was cultured in incubator for 2 hours. After cells were labeled, culture medium was removed and cells were washed 1–2 times with PBS for 3–5 minutes each time. The cell fixing solution (500 μL of 4% paraformaldehyde) was added and incubated for 30 minutes at room temperature. The cells were washed 2 times with PBS for 3–5 minutes each. 50 μL Glycine solutions (2 mg/mL) was added and shaking table was incubated at room temperature for 5 minutes. After removing the glycine solution, the cells were rinsed 1–2 times for 3–5 minutes each time using PBS. 400 μL of penetrant was added, and shaking the table was incubated for 10 minutes, and cells were washed with PBS for 3–5 minutes. The Apollo 567 stains (400 μL) was added and shaking table was incubated for 30 minutes (avoid light, room temperature), Apollo medium was removed 2 times with 400 μL of penetrant (0.5% Triton X-100 in PBS), shaking table was incubated at room temperature for 10 minutes. DAPI was added for nuclear staining for 30 minutes, DAPI was removed, and cells were washed 1–3 times with PBS for 10 minutes each. A fluorescent microscope (Motic, Ximen, China) was used to observe the cells in the dark, and images were captured.

## Oil red O staining and quantification

After the cells were cultivated to the point of final differentiation, the liquid was removed. Adipocytes were washed 3 times with PBS, the cell fixation solution (4% paraformaldehyde) was added, and cells were fixed in 37°C incubators for 40–45 minutes. The Oil red O solution was added for 45 minutes (Oil red O covers the bottom). Cells were washed. Staining was observed under a microscope, and pictures were taken. Oil Red O dye was dissolved with isopropyl alcohol. Absorbance was analyzed at 510 nm wavelength.

## Immunofluorescence staining

GPX2 adenovirus (2 μL) was added when cells grew to 70% density. After the cells converged, DMEM/F12 containing 4% horse serum and 100 units/mL penicillin-streptomycin was used for myogenic differentiation. The cells were collected after a certain time of induction. After adding the cell fixing solution (4% paraformaldehyde) and letting it sit at room temperature for 15 minutes, the cells were three times rinsed with PBS for five minutes each. Permeability with 0.5% Triton X-100 for 10 minutes and cells were washed 3 times. Cell nuclear was stained

through DAPI for 25 minutes at room temperature. Fluorescence was observed with a Motic fluorescence microscope (Motic, Ximen, China). Because the virus overexpression vector contains green fluorescent protein, which fills the muscle fiber. Therefore, the muscle differentiation index and myotube fusion index can be calculated directly after nuclear staining.

## Quantitative real-time PCR (qRT-PCR)

Glyceraldehyde-3-phosphate dehydrogenase (*GAPDH*) was used as an internal reference for all gene normalization analyses. The PCR reaction system used 15 μL containing 5.5 μL RNase-free $H_2O$, 1 μL of cDNA (Concentration 100–200 ng/μL), 7.5 μL SYBR Select Master Mix (ABI, USA), and 0.5 μL of forward and reverse primers (10 μmol/mL). Each qRT-PCR cycle was conducted as follows: 95˚C for 5 minutes; 36 cycles at 95˚C for 10 s and 60˚C for 1 minute. Each experiment was set up with 3 biological replicates. The $2^{-\Delta\Delta Ct}$ method was used to calculate the relative gene expression levels. The expression of genes was detected by qRT-PCR with the same methods. The primer sequences are in S1 Table.

## Western blotting analysis

The samples were ground in the ice bath. RIPA buffer (Thermo Fisher Scientific) was added containing protease inhibitors and phosphatase inhibitors, lysis was on ice for 3–5 minutes, and the cell solution was collected in a centrifuge tube. Set aside on ice for 10 minutes and was centrifuged at 12,000 rpm/min for 10 minutes at 4˚C in an ultracentrifuge. Supernatant was transferred to another centrifuge tube. According to the ratio, 5 × protein loading buffer was added, boiled in hot water at 100˚C for 10 minutes, and divided and the protein samples was stored at -80˚C. For Western blot detection, the protein samples and markers were separated through SDS-PAGE gel electrophoresis and then transferred to PVDF membranes (Millipore, Bedford, MA, USA). A large amount of heat was generated during the transfer and an ice bath was required. After membrane transfer, PVDF membrane was blocked in 5% skim milk for 2 hours and incubated at 4˚C overnight with primary antibodies. The primary antibody was diluted according to instructions. Then PVDF membrane was washed 3 times with TBST for 10 minutes each time. The secondary antibody was incubated at room temperature for 2 hours and washed 3 times with TBST for 10 minutes each time. The dilution ratio of the secondary antibody horseradish peroxidase was 1:10000. Chemiluminescence method was used, and finally, the target protein was detected by a gel imaging system, and protein grayscale values for the protein strip image was analyzed through Image J software.

## Oxidative stress model

After adding $H_2O_2$, the cell was cultured for an additional four hours. The final concentration of $H_2O_2$ was 100 μmol/L. Culture medium was removed and cells were washed 2 times with PBS. It was incubated in 37˚C and 5% $CO_2$ incubator for 24 hours.

## Statistical analysis

The experiments were set up with three biological replicates. The qRT-PCR results were analyzed by the $2^{-\Delta\Delta Ct}$ method. The proteins were analyzed in grayscale using Image J software to obtain quantitative data. The qRT-PCR, proteins quantitative, and other data results were obtained in the experiment were subjected to an independent sample t-test and one-way analysis of variance using SPSS 25 software, and differences were carried out. $p < 0.001$ and $p < 0.01$ indicates a highly significant difference, $p < 0.05$ indicates a statistically significant

difference, and p > 0.05 indicates a non-significant difference. Test data are presented as mean ± standard deviation.

## Results

### Overexpression of *GPX2* inhibits proliferation of porcine preadipocytes

Porcine preadipocytes were subjected to adenovirus overexpression of *GPX2* in order to clarify the impact of the proliferative state. This research found that GPX2 mRNA ($P < 0.01$) and protein ($P < 0.05$) were successfully overexpressed in porcine preadipocytes (Fig 1A and 1D).

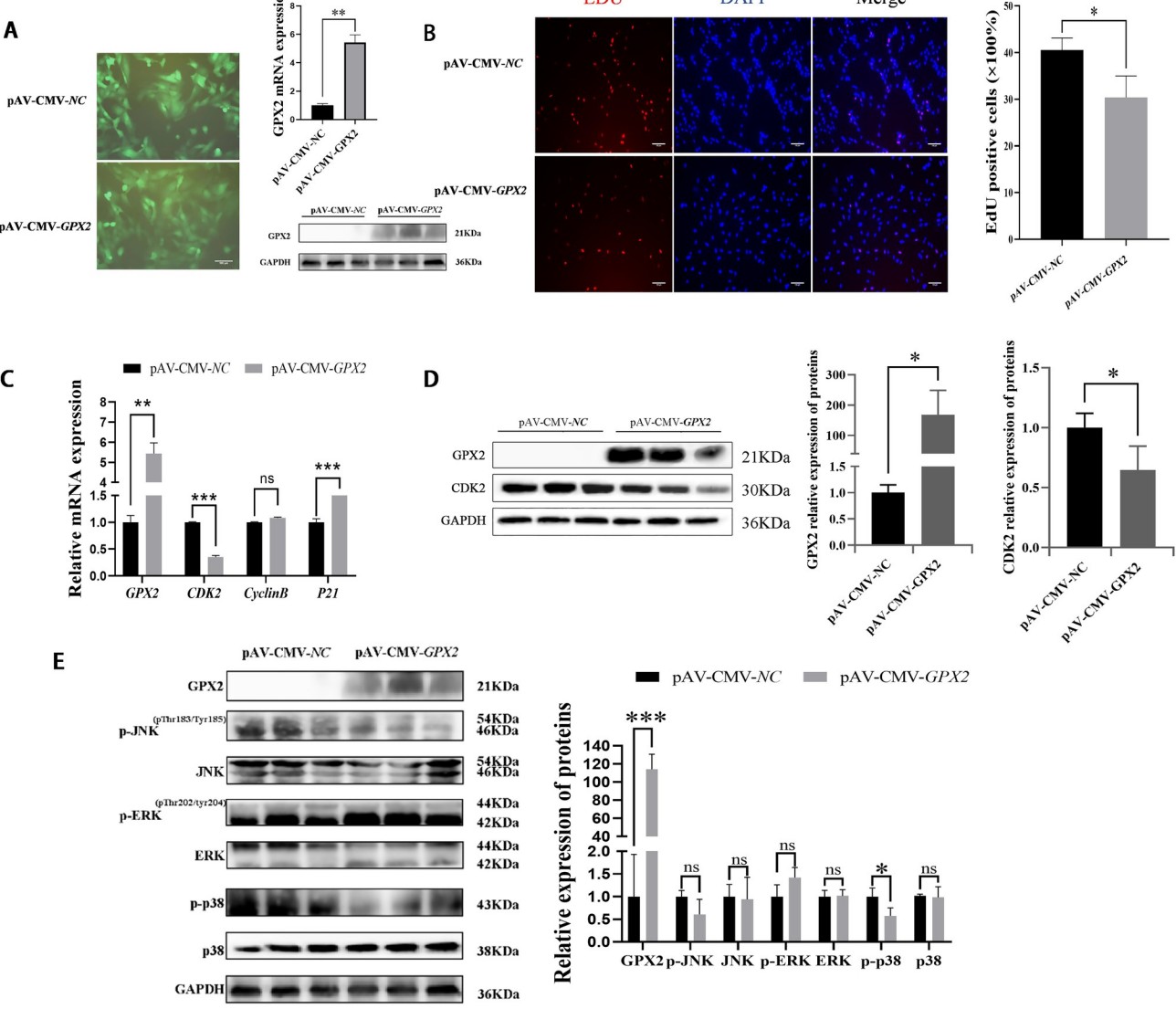

**Fig 1. Overexpression of *GPX2* inhibits proliferation of porcine preadipocytes.** (A) Porcine preadipocytes were infected with adenovirus carrying GPX2, and the mRNA and protein of GPX2 stable expressed in porcine preadipocytes. (B) Representative immunostaining images comparing the cells in proliferative phase from negative control and overexpression *GPX2*, the nuclei are stained with DAPI. Quantification graph showing percentage of cells positive for the red cell (EdU). (C) mRNA expression of proliferation-related gene between overexpression *GPX2* and negative control group. (D) Protein levels and abundance analysis for GPX2 and CDK2 at 48 hours after overexpression of *GPX2* in porcine preadipocytes. (E) Protein levels and abundance analysis of MAPK signaling pathways in proliferation procession. Scale bar = 100 μm, * $P < 0.05$, ** $P < 0.01$, *** $P < 0.001$, ns indicates not significant.

After overexpression of *GPX2*, the proliferation of preadipocytes was detected by EdU staining, and we found that the proliferation rate of porcine preadipocytes was significantly reduced (22%) ($P < 0.05$) (Fig 1B). The mRNA and protein expressions of cell cycle-related genes were detected through qRT-PCR and Western blotting. The mRNA expression of the *P21* was higher in the overexpression group than in the control group ($P < 0.001$). And mRNA ($P < 0.001$) and protein ($P < 0.05$) expressions of CDK2 were decreased (Fig 1C and 1D). In porcine preadipocytes, the protein levels of p-JNK, JNK, p-ERK1/2, ERK1/2, p-p38, and p38 were examined in order to comprehend the impact of *GPX2* on the MAPK pathway. The result illustrates that p-JNK and p38 protein levels were restrained, but no significant difference after *GPX2* overexpression ($P > 0.05$), and the protein expression of p-p38 was down-regulated ($P < 0.05$) (Fig 1E). In conclusion, overexpression of the *GPX2* gene could reduce the proliferation ability of porcine preadipocytes, and the p-p38 was suppressed.

### *GPX2* promoted lipid degradation and inhibit adipogenic differentiation of porcine preadipocytes

Adipogenesis was assessed on day 6. The lipid content of porcine preadipocytes was decreased after overexpression of *GPX2*, as assessed by Oil red O staining ($P < 0.01$) (Fig 2A). Quantification of lipid accumulation indicated that differentiation of porcine preadipocytes was very significantly inhibited by overexpression of *GPX2*, and the mRNA of adipogenesis and lipid degradation-related genes were analyzed (Fig 2B and 2D). The results showed that the mRNA level of CCAAT/enhancer binding protein α (*CEBPα*) was increased ($P < 0.05$) after *GPX2* overexpression, and peroxisome proliferator-activated receptor γ (*PPARγ*) was reduced but not at a significant level ($P > 0.05$) (Fig 2B). The fatty acid synthase (FAS) ($P < 0.05$) and CEBPα ($P < 0.05$) protein expression were dropped in the overexpression of GPX2 than in the control group (Fig 2C), and mRNA ($P < 0.001$) and protein ($P < 0.05$) levels of AP2 were significantly up-regulated (Fig 2B and 2C). In addition, the mRNA expression level of triacylglyceride lipase (*ATGL*) and lipoprotein lipase (*LPL*) genes was an increasing trend in overexpression of *GPX2* ($P < 0.01$) (Fig 2D). And the hormone-sensitive lipase (HSL) also was raised ($P < 0.01$) (Fig 2E).

The phosphofructokinase-muscle (*PFKM*) ($P < 0.05$) and protein kinase AMP-activated non-catalytic subunit gamma 3 (*PRKAG3*) ($P < 0.01$) in the overexpression group was higher than in the control group, but pyruvate kinase-muscle (*PKM*) was significantly reduced ($P < 0.001$) (Fig 3A). The protein level of PKM also dropped ($P < 0.01$) (Fig 3B), which is consistent with the results of mRNA. In addition, the change in the MAPK pathway also was noticed after overexpression with *GPX2*. JNK and p38 protein expression were reduced in adipogenic differentiation, but no significant difference ($P > 0.05$). And ERK was significantly decreased ($P < 0.01$), the p-p38 was increased ($P < 0.01$) (Fig 4).

### Overexpression of *GPX2* induces the proliferation of porcine skeletal muscle

Adenovirus overexpression of *GPX2* in porcine skeletal muscle cells was used to access specificity in proliferation situations. GPX2 mRNA ($P < 0.001$) and protein ($P < 0.01$) were successfully overexpressed in porcine skeletal muscle cells (Fig 5A). The cell proliferation was detected by EdU staining. The proliferation rate was significantly increased (67%) by overexpression of *GPX2* ($P < 0.001$) (Fig 5B), and the mRNA level of *CDK2* was not significantly changed ($p > 0.05$), but protein expression was raised ($P < 0.05$) (Fig 5C and 5D). In addition, the mRNA of *CyclinB* also were up regulated. It had dropped on the mRNA of the *P21* (Fig 5C). After overexpressing *GPX2*, c-fos protein expression has a declining trend ($P > 0.05$).

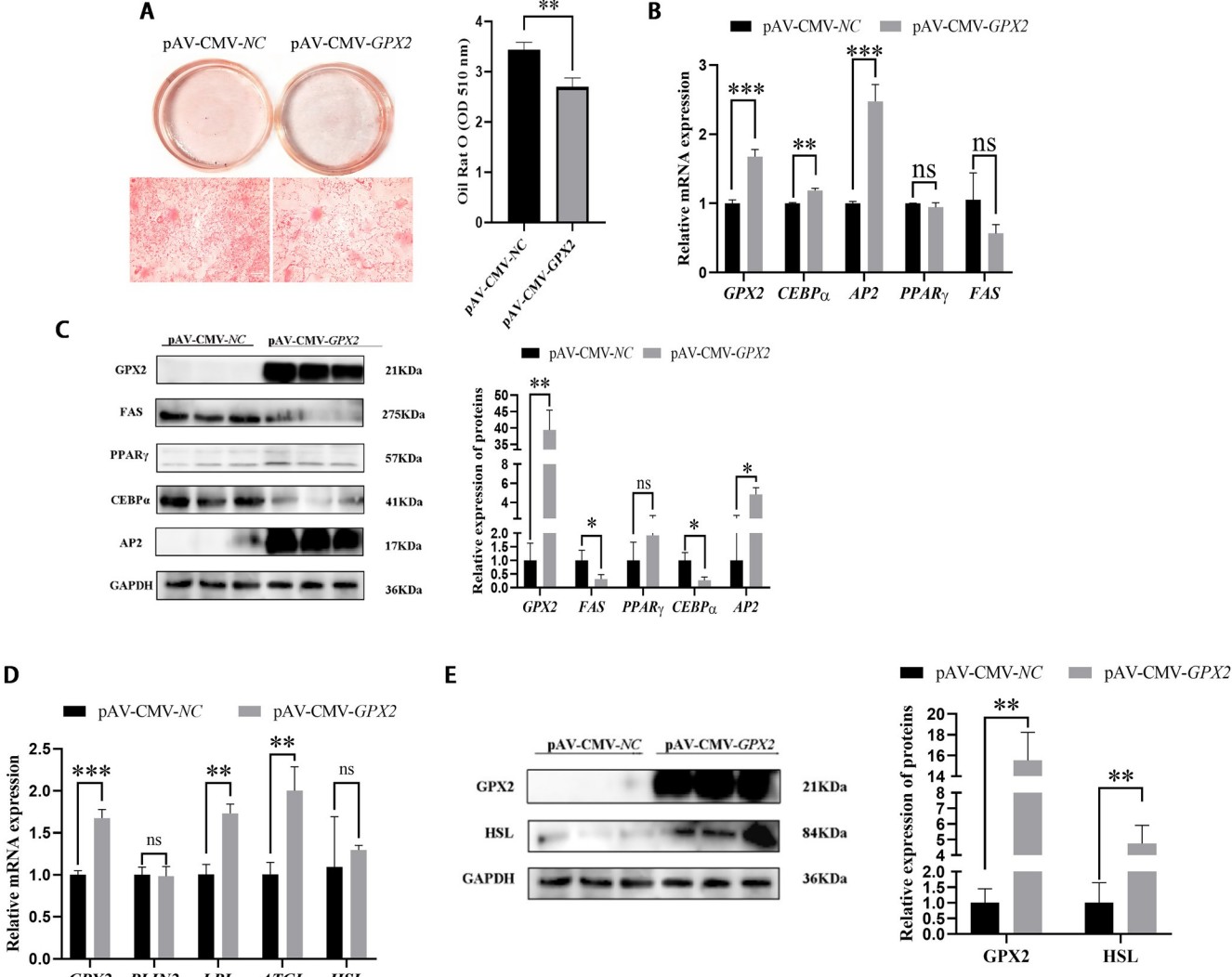

**Fig 2. Overexpression of *GPX2* inhibits adipogenic differentiation and promoted the lipid degradation of porcine preadipocytes.** (A) Photographs and micrographs of porcine preadipocytes stained with Oil red O, and detection of lipid contents from negative control and overexpression of *GPX2*. (B) mRNA expression of adipogenic regulator of porcine preadipocytes after 6 days' differentiation. (C) Protein expression and abundance analysis for adipogenic regulator in porcine preadipocytes after 6 days' differentiation. (D) mRNA expression of lipid degradation in control group and overexpression of *GPX2* group of porcine preadipocytes after 6 days' differentiation. (E) Protein levels and abundance analysis for GPX2 and HSL after overexpression of *GPX2* in porcine preadipocytes. Scale bar = 100 μm, * $P < 0.05$, ** $P < 0.01$, *** $P < 0.001$, ns indicates not significant.

And JNK ($P < 0.05$), ERK1/2 ($P < 0.05$), and p-p38 ($P < 0.05$) levels were increased (Fig 5E). These results showed that the JNK, ERK1/2, and p38 pathways were activated, and that overexpression of the *GPX2* gene might increase the capacity of pig skeletal muscle cells to proliferate.

## Overexpression *GPX2* promoted the myoblastic differentiation of porcine skeletal muscle

To ascertain how *GPX2* affects myoblastic differentiation. Porcine skeletal muscle cells were overexpressed for *GPX2*, and then myoblast differentiation was induced. Using GFP in an adenovirus overexpression vector, the cell contour was seen, and DAPI was used to label the

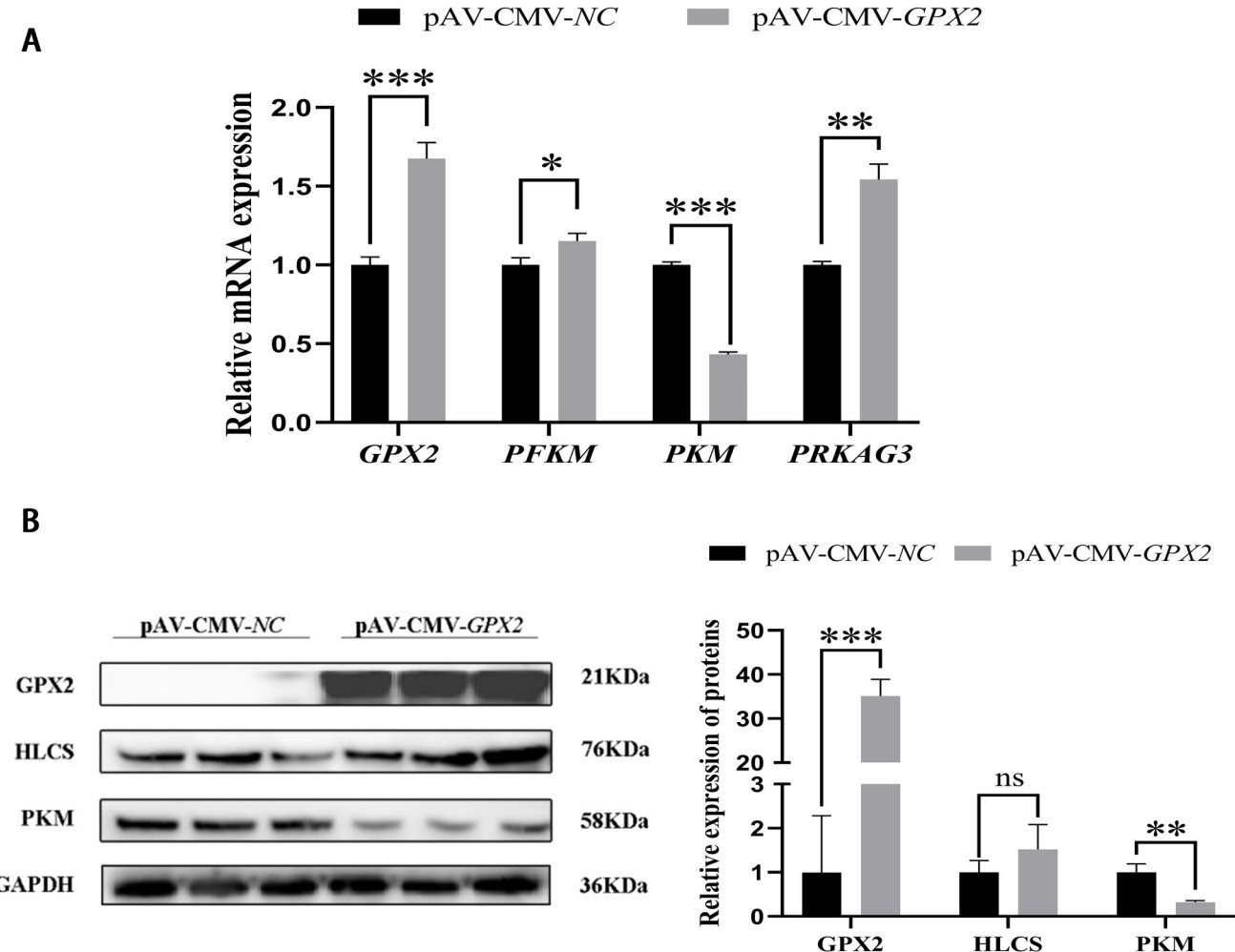

**Fig 3. Overexpression of *GPX2* promotes glycolysis-related genes expression of porcine preadipocytes.** (A) mRNA expression of glycolytic-related genes of porcine preadipocytes. (B) Protein expressions and abundance analysis for HLCS and PKM in porcine preadipocytes from negative control and overexpression of *GPX2*. * $P < 0.05$, ** $P < 0.01$, *** $P < 0.001$, ns indicates not significant.

nucleus. The porcine skeletal muscle cells overexpressing *GPX2* had an increased effect on the differentiation index and fusion index, but no significant difference ($P > 0.05$) (Fig 6A). Compared with the negative control group, mRNA expression of myogenin (*MYOG*) ($P < 0.001$), myogenic differentiation 1 (*MYOD*) ($P < 0.01$), and myosin heavy chain 3 (*MYH3*) ($P < 0.001$) were very significantly increased after overexpression of *GPX2*, and myostatin (*MSTN*) mRNA level has significantly dropped ($P < 0.01$) (Fig 6B). The protein levels of MYH3 and MYOG in *GPX2* overexpression group were higher than in the normal group ($P < 0.05$) (Fig 6C). The *PFKM* and *PKM* mRNA levels were reduced after overexpression of *GPX2* ($P < 0.01$) (Fig 7A). And the protein expression of HLCS and PKM also decreased ($P < 0.05$) (Fig 7B). After *GPX2* overexpression, the JNK protein expression has a down-regulated trend ($P > 0.05$), and p-ERK1/2 and ERK1/2 were significantly increased ($P < 0.05$), but the protein level of p38 was dropped ($P < 0.05$) (Fig 8). When considered collectively, our findings demonstrated that *GPX2* gene overexpression stimulates myoblastic differentiation of pig skeletal muscle cells, and that the control of this process may be mediated via the MAPK signaling system.

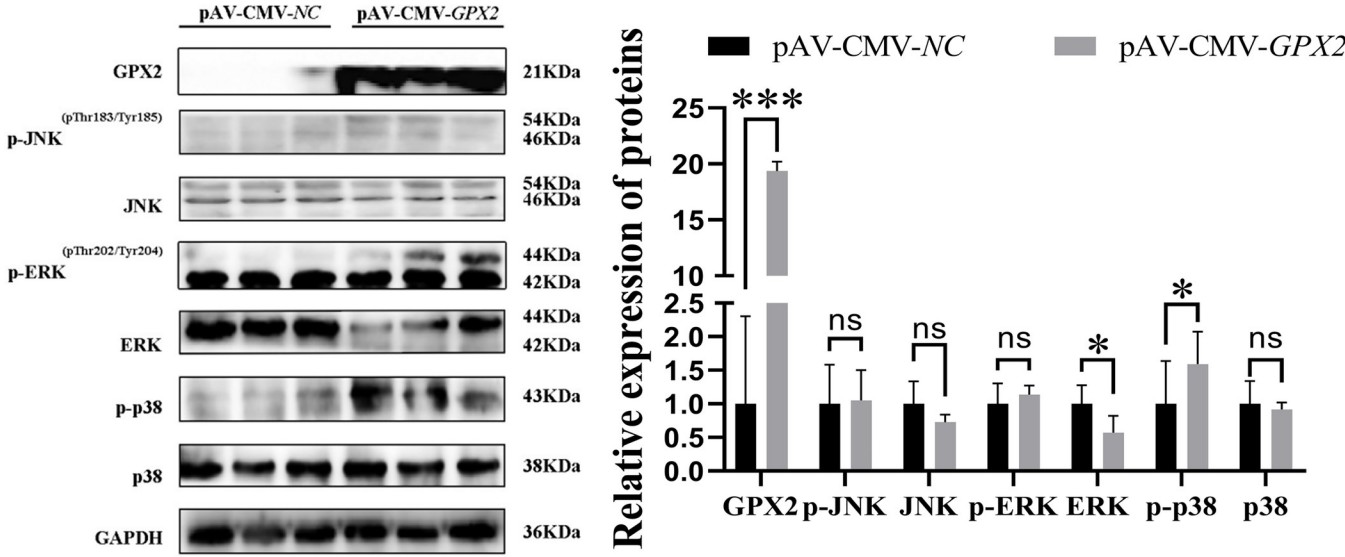

**Fig 4. The change of MAPK signaling pathway after overexpression with *GPX2*.** Protein levels and abundance analysis of MAPK signaling pathway in adipogenic differentiation. * $P < 0.05$, ** $P < 0.01$, *** $P < 0.001$, ns indicates not significant.

## Discussion

Numerous studies have demonstrated the ability of GPX2 to control cell division and proliferation [10,12,13]. However, the majority of GPX2 cell proliferation research being done now focuses on cancer and other issues [14,15], and the effects on porcine adipocytes and skeletal muscle cells have not been reported. Our research revealed that the overexpression of *GPX2* suppressed the growth of porcine preadipocytes. The cell cycle consists of four distinct phases, including G1 phase, S phase, G2 phase, and M phase. CDK2 and P21 have been reported to regulate cell cycle progression [16,17], and CDK2 was required for G1 phase progression and entry into the S phase [18]. Previous research has found that cell cycle progression and DNA synthesis were inhibited after CDK2 activity was suppressed [19]. According to two studies, P21 directly upstream regulates CDK2 activity, a key factor that inhibits cell growth [20,21]. These findings support our hypothesis that, following *GPX2* overexpression, CDK2 activity was decreased and porcine preadipocyte proliferation was subsequently suppressed. To the best of our knowledge, these results supported the critical function of *GPX2* in controlling the proliferation of porcine preadipocytes. Research has found that ERK, JNK, and p-p38 plays an important role in cell proliferation [22–24]. Recent studies showed that activation of ERK was necessary for preadipocyte proliferation, and ERK inhibitors had anti-proliferative activity [25,26]. It has been demonstrated that p-p38 inhibition inhibited TGF-β1-induced proliferation and decreased DNA synthesis [27]. Our results show that p-p38 protein expression levels were decreased after overexpression of *GPX2*. Therefore, we suspect that proliferation of porcine preadipocytes was inhibited, probably because *GPX2* impeded DNA synthesis by inhibiting p-p38, which in turn inhibited cell cycle progression, but the specific mechanism is required for further studies.

Prior research has emphasized the significance of PPARγ and C/EBPα in controlling adipogenesis, with their transcriptional activation serving as the first trigger for adipogenesis [28,29]. According to our findings, adipogenic differentiation was suppressed and PPARγ and C/EBPα protein levels were downregulated following *GPX2* overexpression. In addition, we found that the mRNA and protein expression of AP2 was increased, which may be because

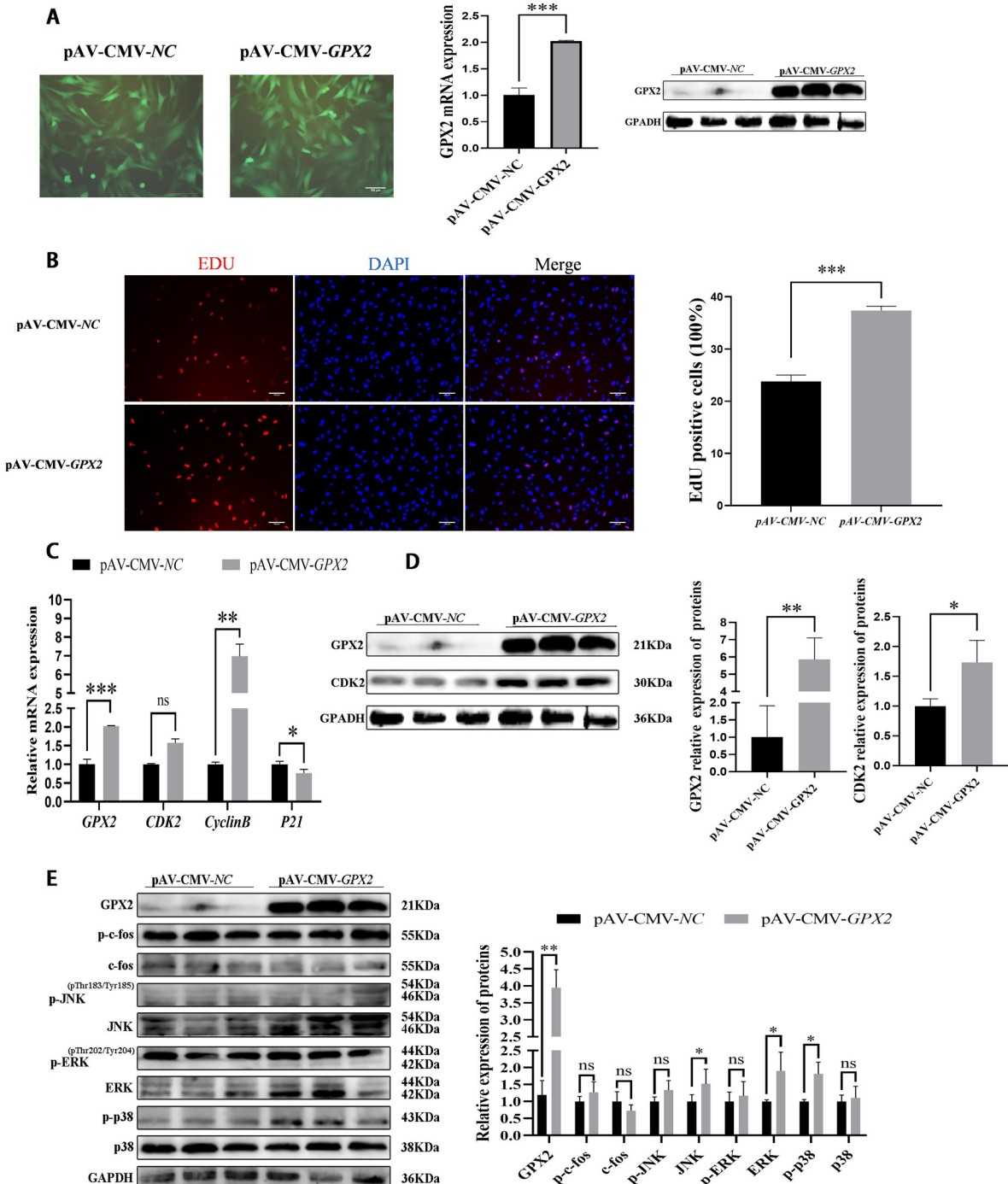

**Fig 5. Overexpression of *GPX2* promotes proliferation of porcine skeletal muscle cells.** (A) Porcine skeletal muscle cells were infected with adenovirus carrying *GPX2*, and the mRNA and protein of GPX2 were stably expressed in the porcine skeletal muscle cells. (B) Representative immunostaining images comparing the cells in proliferative phase from negative control and overexpression of *GPX2*, the nuclei was stained with DAPI. Quantification graph showing percentage of cells positive for the red cell (EdU). (C) mRNA expression of proliferation-related gene between overexpression of *GPX2* and control group. (D) Protein levels and abundance analysis for GPX2 and CDK2 at 48h after overexpression of GPX2 in the porcine skeletal muscle cells. (E) Protein levels and abundance analysis of MAPK signaling pathway in proliferation stage after overexpression of *GPX2* in porcine skeletal muscle cells. Scale bar = 100 μm, $^*$ $P < 0.05$, $^{**}$ $P < 0.01$, $^{***}$ $P < 0.001$, ns indicates not significant.

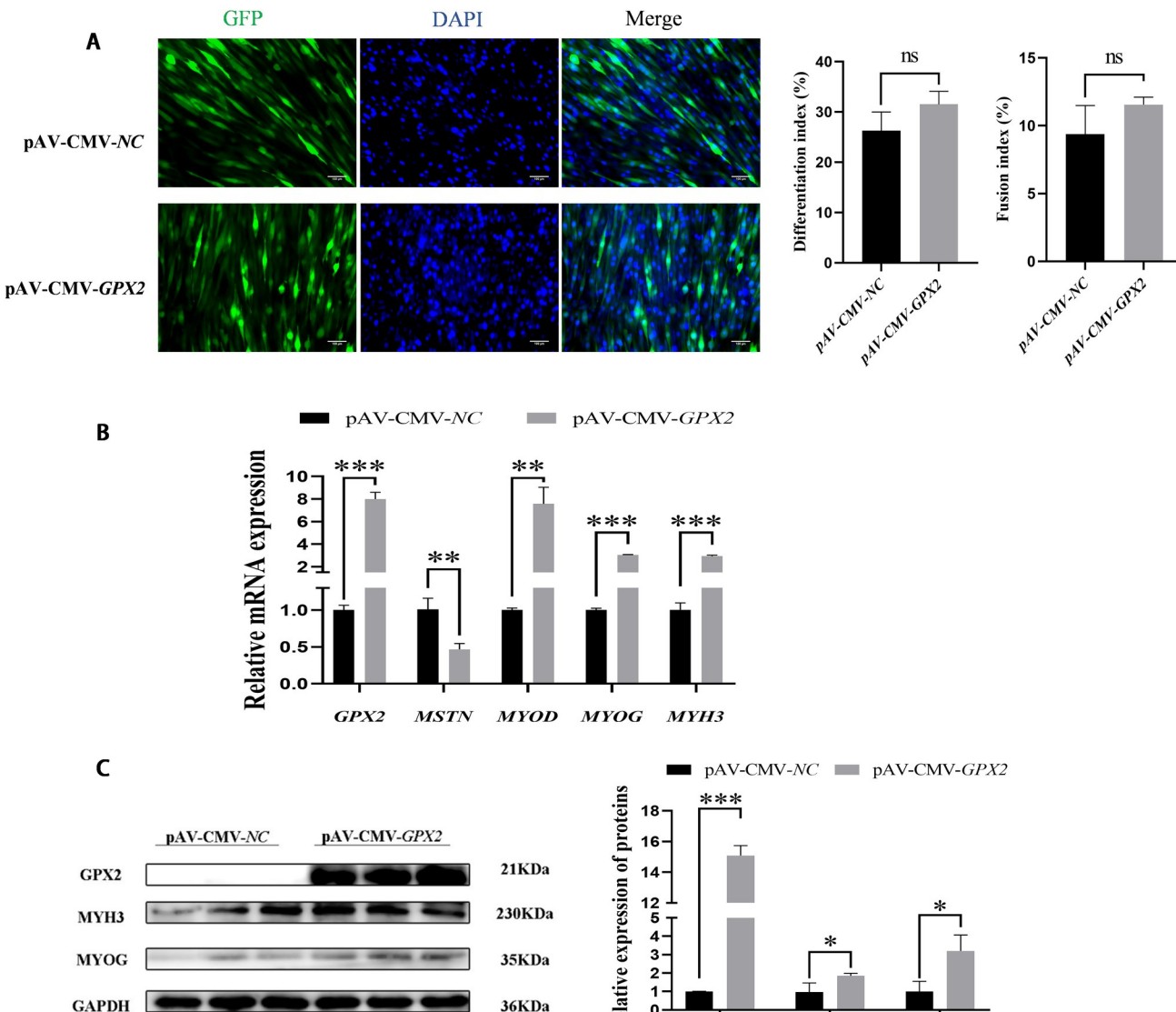

**Fig 6. Overexpression of *GPX2* promotes myoblastic differentiation of porcine skeletal muscle cells.** (A) Representative immunostaining images comparing the cells in the myogenic differentiation stage from negative control and overexpression of *GPX2*. Blue represents nuclear staining; Green represents green fluorescent protein. The differentiation index was calculated as the percentage of the number of nuclei in the myotube to the total nucleus. The fusion index was calculated as the percentage of the number of myotubes with more than two nuclei in the total nucleus. (B) The mRNA expression of myogenic-related factors. (C) Protein levels and abundance analysis of myogenic marker protein after overexpression of *GPX2* in porcine skeletal muscle cells. Scale bar = 100 μm, * $P < 0.05$, ** $P < 0.01$, *** $P < 0.001$, ns indicates not significant.

lipid degradation was promoted after overexpression of *GPX2*. Studies have shown that expression of AP2 was increased under the agents of stimulated lipolysis in both mouse and human adipocytes [30]. We also investigated the fascinating connection between GPX2 and lipid breakdown in this work. Our results are in line with these findings that lipolysis markers were raised after overexpression of *GPX2* [31–34], such as *ATGL*, *LPL*, and *HSL*. PFKM and PKM could catalyze rate-limiting steps of glycolysis. The accumulation of TG precursors was increased via glycolysis. It's interesting to note that our findings show that *PFKM* and *PRKAG3* increased following *GPX2* overexpression. To understand the chemical mechanism behind this phenomena, more research is required. Some studies showed that ROS could

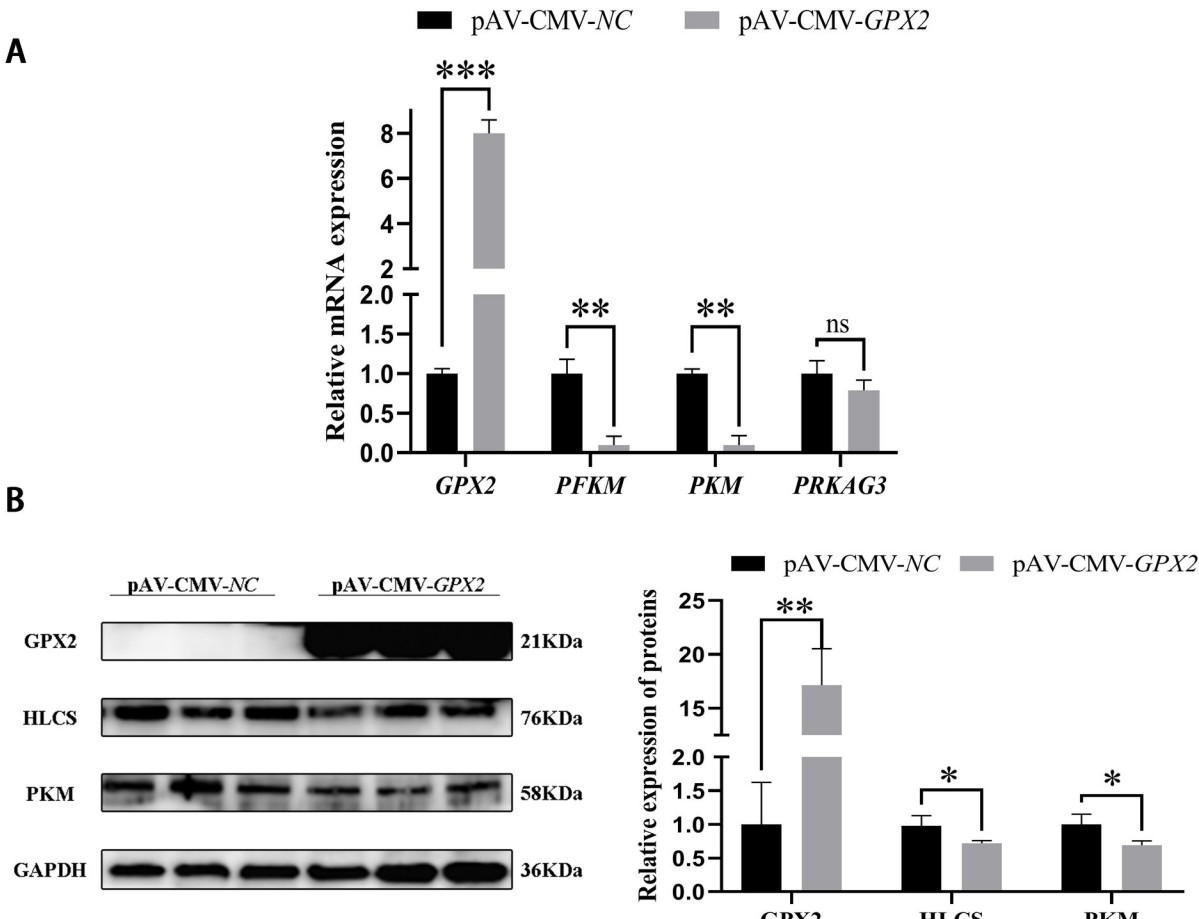

**Fig 7. Overexpression of *GPX2* inhibits glycolysis of porcine skeletal muscle cells.** (A) mRNA expression of glycolytic-related genes in porcine skeletal muscle cells. (B) Protein expressions and abundance analysis of HLCS and PKM in porcine skeletal muscle cells from negative control and overexpression of *GPX2*. * $P < 0.05$, ** $P < 0.01$, *** $P < 0.001$, ns indicates not significant.

promote differentiation of 3T3-L1 preadipocytes [35]. Crucially, lowering ROS in developed cells prevented adipogenic development and lipid accumulation [36,37], and lowering ROS levels in cells following *GPX2* overexpression. These results suggested that adipocytes were affected by antioxidation of *GPX2*. Another antioxidant that decreases lipid production and increases lipolysis produced comparable effects in other animals or cells [38,39].

Modulation of the MAPK signaling pathway plays a critical role in differentiation of adipogenesis [40]. ERK was necessary for PPARγ and CEBPα genes transcription, and the adipocyte differentiation was promoted via activation of ERK1/2 pathway [41,42]. The study discovered that when the ERK1/2 pathway was blocked, the quantity of fat cells was dramatically decreased [43]. However, p-p38 was suppressed during the process of 3T3-L1 adipogenic differentiation was inhibited [44]. The results of this study were opposite. We have found that p-p38 was up-regulated in process of adipogenic differentiation, and further studies are needed in subsequent experiments. In this study, we also found that ERK1/2 expression was decreased during adipogenic differentiation. These results suggested that *GPX2* overexpression may inhibit differentiation of porcine preadipocytes through ERK1/2 and p38 signaling pathways. Although cellular oxidative stress resulted in a highly significant increase in GPX2 and NRF2 proteins (S1 Fig). However, overexpression of *GPX2* did not result in a change in NRF2

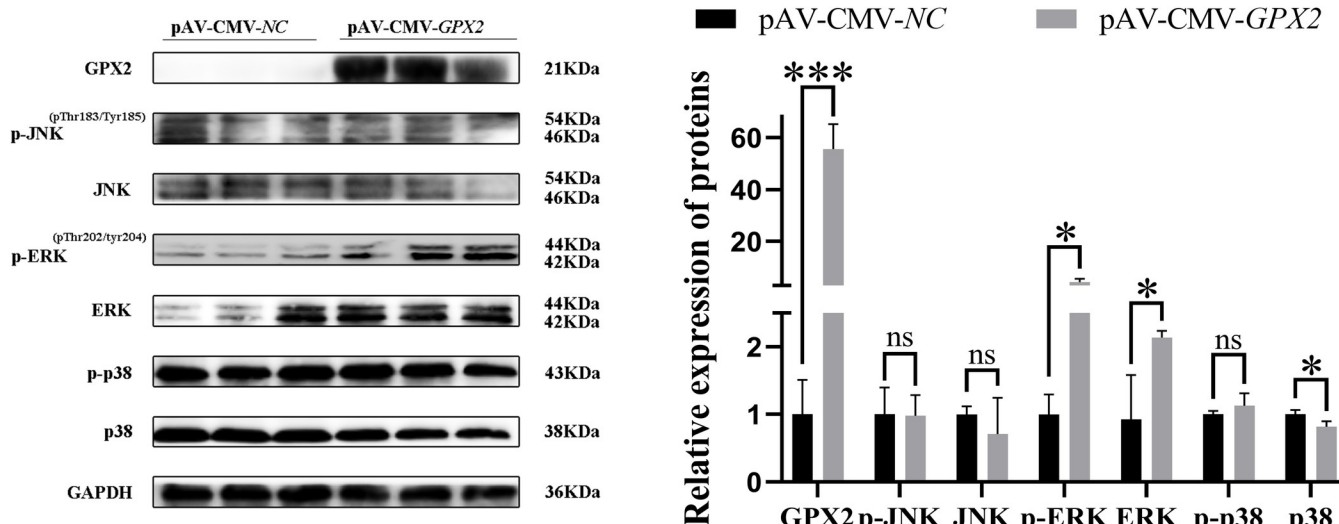

**Fig 8. The changes of MAPK signaling pathway after overexpression of *GPX2*.** Protein levels and abundance analysis of MAPK signaling pathway after overexpression *GPX2* in myoblastic differentiation stage. * $P < 0.05$, ** $P < 0.01$, *** $P < 0.001$, ns indicates not significant.

protein expression during both the proliferation and differentiation stages of porcine adipocytes (S2 Fig). In other words, overexpression of *GPX2* causes modifications in the protein expression of the MAPK signaling pathway, independent of NRF2.

Skeletal muscle's capacity to multiply has an impact on the number of muscle cells. According to some research, oxidative damage can prevent cell division by encouraging the expression of the *P53* and *P21* genes in pancreatic islet and human kidney epithelial cells [45,46]. Recent studies have demonstrated that myoblast proliferation was impaired by oxidative stress [47]. GPX2 can reduce oxidative stress. Our findings imply that *GPX2* overexpression stimulated the growth of skeletal muscle cells in pigs. In this study, we found that overexpression of *GPX2* increases JNK, ERK, and p-p38 expression in proliferation process of porcine skeletal muscle cells. Several studies have reported that cell proliferation was promoted via activation of JNK [48,49]. Similar to this, it has been shown that ERK activation mediates cell proliferation and cell cycle progression in myoblasts [50], and early ERK signal was a crucial factor in determining the proliferation of muscle stem cells [51]. These results suggested that the proliferation of porcine skeletal muscle cells was regulated by MAPK signaling pathway after overexpression of *GPX2*.

Recent research has shown that the marker genes *MYOD*, *MYOG*, and *MYH3* controlled the differentiation of primary myoblasts [52]. As additional research comes to light, it is becoming increasingly clear that MYOD played a crucial role in myogenic differentiation, which promotes myogenin expression and eases myocyte fusion [53]. The mitochondrial-related metabolic functions can be controlled by MYOD, which was necessary for myogenic differentiation [54]. According to recent research, skeletal muscle cells' ability and differentiation factors were diminished because of Se deficit [55]. In this study, we found that the myoblastic differentiation was promoted after overexpression of *GPX2*. These results suggested that via controlling mitochondrial activities, *GPX2* may encourage the development of skeletal muscle cells in pigs. In our present study, we confirmed that PFKM, PKM, and PRKAG3 are decreased in myoblastic differentiation of porcine skeletal muscle cells. This might be the result of a reduction in myotube fusion during the last stages of swine skeletal muscle cell development, which lowered the fusion process's energy needs.

Some studies demonstrated that JNK is the key player during skeletal muscle development. Muscle differentiation was negatively regulated through the JNK/MAPK signaling pathway [56,57]. Additionally, ERK 1/2 activation on skeletal muscle cells accelerated myoblast differentiation [58]. This phenomenon is in line with our results. Research has demonstrated that myoblast differentiation was caused by the activation of p38 MAPK [59]. Further research is required to identify the mechanism behind the observed considerable drop in the p38 pathway and rise in the ERK1/2 pathway during the myoblastic differentiation of pig skeletal muscle cells. These results demonstrate that p38 MAPK pathway regulation of myoblastic differentiation of porcine skeletal muscle cells followed *GPX2* overexpression. The expression of NRF2 protein did not change after overexpression of *GPX2* in the proliferative or differentiated stage of porcine skeletal muscle cells (S3 Fig). Put differently, NRF2 did not mediate the alteration in the protein expression of the MAPK signaling pathway produced by *GPX2* overexpression; rather, other processes were involved. To put it briefly, *GPX2* encourages the growth of muscle and prevents the accumulation of fat. Furthermore, our earlier research discovered that the combination genotype G3 increased the expression of *GPX2* in several organs [10]. Thus, selecting people with high *GPX2* genes may be accomplished by using genetic approaches to identify the SNPs boosting *GPX2* expression. In conclusion, it has been shown that *GPX2* controls the proliferation and differentiation of skeletal muscle cells and porcine preadipocytes through MAPK pathways (Fig 9). Following *GPX2* overexpression and p38 pathway suppression, the proliferation of porcine preadipocytes was decreased. Lipid breakdown was encouraged and adipogenic differentiation was prevented, whereas p-p38 was elevated and ERK1/2 was decreased. Activation of JNK, ERK1/2, and p-p38 together with overexpression of *GPX2* resulted in the proliferation of skeletal muscle cells in pigs. It encouraged the development of

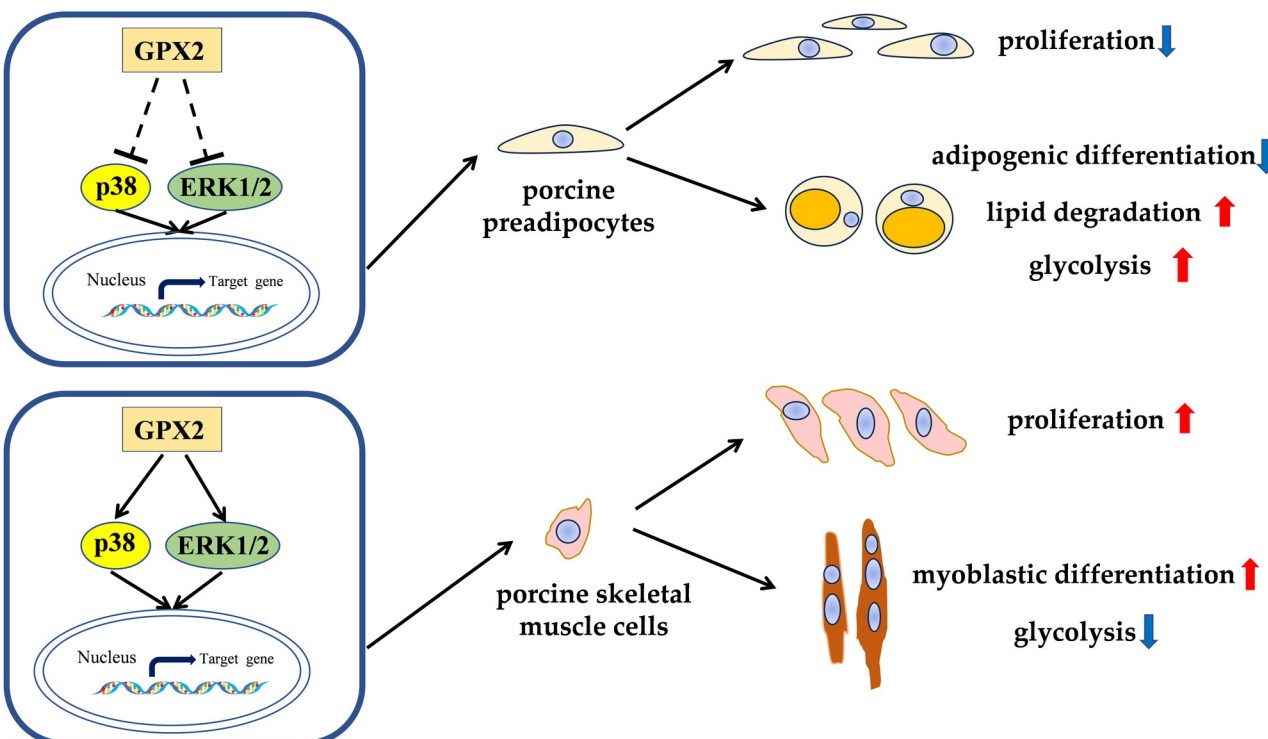

**Fig 9. Hypothesis schematic models illustrating the *GPX2* affects the proliferation and differentiation of porcine preadipocytes and skeletal muscle cells by regulating MAPK pathways.** ↑ indicates expression up-regulation, ↓ indicates expression down-regulation.

myoblasts. Myoblastic differentiation involved the activation of the ERK1/2 pathway and the suppression of p38 activation. According to the aforementioned findings, the *GPX2* gene can control the MAPK signaling system to lower animal body fat and increase muscle development.

## Supporting information

**S1 Table. Primer information.** Primers for RT-PCR.
(DOCX)

**S1 Fig. GPX2 and NRF2 expression were increased after oxidative stress.**
(TIF)

**S2 Fig. Overexpression of *GPX2* did not result in a change in NRF2 protein expression in porcine preadipocytes.** (A) Protein expressions and abundance analysis of GPX2 and NRF2 in the proliferative stage of porcine preadipocytes after overexpression of *GPX2*. (B) Protein expressions and abundance analysis of GPX2 and NRF2 in the differentiated stage of porcine preadipocytes. * $p < 0.05$, ** $p < 0.01$, *** $p < 0.001$, ns indicates not significant.
(TIF)

**S3 Fig. Overexpression of *GPX2* did not result in a change in NRF2 protein expression in porcine skeletal muscle cells.** (A) Protein expressions of GPX2 and NRF2 in the proliferative stage of porcine skeletal muscle cells after overexpression of *GPX2*. (B) Protein expressions of GPX2 and NRF2 in the differentiated stage of porcine skeletal muscle cells.
(TIF)

**S1 File.**
(PDF)

## Author Contributions

**Conceptualization:** Lei Wang, Lei Pu.

**Data curation:** Lei Qin, Yunyan Luo, Zuochen Wen.

**Formal analysis:** Chunguang Zhang, Lei Pu.

**Funding acquisition:** Guofang Wu, Lei Pu.

**Investigation:** Chunguang Zhang, Lei Qin, Shifan Deng, Lei Pu.

**Methodology:** Xin Liu, Lei Pu.

**Project administration:** Lei Wang, Xin Liu.

**Resources:** Lei Wang.

**Supervision:** Yuhong Ma.

**Validation:** Lei Wang, Xin Liu, Lei Pu.

**Visualization:** Jianbo Zhang, Chao Sun, Lei Pu.

**Writing – original draft:** Chunguang Zhang, Lei Qin.

**Writing – review & editing:** Akpaca Samson Vignon, Chunting Zheng, Xueli Zhu, Han Chu, Liang Hong, Jianbin Zhang, Hua Yang.

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
