## [Decision Letter · Decision Letter 0]

6 Sep 2023

PONE-D-23-26748Overexpression of GPX2 gene regulates the development of porcine preadipocytes and skeletal muscle cells through MAPK signaling pathwayPLOS ONE

Dear Dr. Pu,

Thank you for submitting your manuscript to PLOS ONE. After careful consideration, we feel that it has merit but does not fully meet PLOS ONE’s publication criteria as it currently stands. Therefore, we invite you to submit a revised version of the manuscript that addresses the points raised during the review process.

We look forward to receiving your revised manuscript.

Kind regards,

Juan J Loor

Academic Editor

PLOS ONE

Journal Requirements:

"Tianjin Natural Science Foundation

(20JCQNJC00650) Lei Pu

Scientific Research Program of Tianjin

Education Commission

(2018KJ188) Lei Pu

Key R&D and Transformation Program of

Qinghai Province Science and

Technology Assistance Program for

Qinghai (2023-NK-141) Guofang Wu

Tianjin pig industry technology system

innovation team (ITTPRS2021006) Jianbin Zhang

National Swine Industry Technology

System (CARS-35) Xin Liu

Agricultural Science and Technology

Innovation Program (ASTIP-IAS02) Not applicable

National undergraduate innovation and

entrepreneurship training program Lei Pu

Tianjin Key Laboratory of Green

ecological feed (TJ202302) Lei Pu

Tianjin excellent agricultural science and

technology special commissioner and

support projects Lei Pu"

7. We note that Figure 9 in your submission contain copyrighted images. All PLOS content is published under the Creative Commons Attribution License (CC BY 4.0), which means that the manuscript, images, and Supporting Information files will be freely available online, and any third party is permitted to access, download, copy, distribute, and use these materials in any way, even commercially, with proper attribution. For more information, see our copyright guidelines: http://journals.plos.org/plosone/s/licenses-and-copyright.

a. You may seek permission from the original copyright holder of Figure 9 to publish the content specifically under the CC BY 4.0 license. 

Reviewers' comments:

Reviewer's Responses to Questions

**Comments to the Author**

1. Is the manuscript technically sound, and do the data support the conclusions?

Reviewer #1: Partly

2. Has the statistical analysis been performed appropriately and rigorously? 

Reviewer #1: Yes

3. Have the authors made all data underlying the findings in their manuscript fully available?

Reviewer #1: Yes

4. Is the manuscript presented in an intelligible fashion and written in standard English?

Reviewer #1: Yes

5. Review Comments to the Author

Reviewer #1: Zhang et al., investigated the role of GPX2 in both adipocytes and myocytes via the overexpression of GPX2 using in vitro models. It provides some insights on the role of GPX2 in regulating lipid metabolism in adipocytes. However, there are some concerns about the study which are listed below:

1. Why did you choose GPX2 not GPX1, GPX3 or GPX4? In other words, what is the difference between GXP1 and the other GPXs? Please clarify this in the introduction. Did you check the expression GPXs in pigs? Is GPX2 expression greater or lower than the other GPXs? Also, did you check the expression of the other GPXs when you overexpressed GPX2 in your cells?

2. Since GPX2 plays critical role in regulating oxidative stress, why did you overexpress it in adipocytes and myocytes not liver? Liver is one of the major sites for ROS production. Have you checked the expression of GPX2 in different tissues in pigs?

3. Why did you choose overexpress of GPX2 rather than knockout or knockdown it?

4. Is there enough evidence suggests that selenium supplementation can increase the expression of GPX2? If not the conclusion about "selenium as a nutritional supplement to promote human subcutaneous

fat reduction and muscle growth" should be revised.

5. Make sure the format is consistent. For example, Line 88: "Porcine skeletal muscle cells culture and myogenic differentiation" is not bold.

6. I feel something is missed between oxidative stress and lipid metabolism. In the introduction you mentioned GPX2 plays an important role in regulating oxidative stress, but how does this lead you to determine whether overexpression of GPX2 may change lipid metabolism? Please make the logic more clear and then readers can understand your story easier.

7. Based on your results, it seems that GXP2 plays different roles in different type of cells. Therefore, you should make careful conclusions that increased expression of GPX2 might be good for human health. Because overexpression of GPX2 in other cells such as hepatocyes or immune cells might not be good for human health. More research is required to further demonstrate the role of GPX2 for overall health in an animal or human.

8. Your results indicate that oxidative stress can increase GPX2. Besides, overexpression of GPX2 seems to be good for health since it inhibit adipocyte proliferation and promotes muscle cell growth. Therefore, do you think mild oxidative stress might be good for health?

Images are not of high quality. Please improve.

6. PLOS authors have the option to publish the peer review history of their article (what does this mean?). If published, this will include your full peer review and any attached files.

Reviewer #1: **Yes: **Yusheng Liang

---

## [Author Response · Author response to Decision Letter 0]

4 Oct 2023

Dear editor：

We appreciate the editor and reviewers very much for their positive and constructive comments and suggestions on our manuscript. We have studied the reviewer’s comments carefully and have made revisions which were marked in yellow in the paper. Furthermore, we have made additions and modifications according to the requirements of the journal. We have uploaded Fig 9 again. It is all drawn by the author, and there is no copyright problem. The original ppt of the manuscript of Fig 9 was shown in the attachment.

Thanks again to the editor and reviewers for their suggestions and comments. If the editorial department has any questions, please do not hesitate to contact us directly.

Sincerely,

Lei Pu 

E-mail: pulei87@126.com

Tianjin Key Laboratory of Agricultural Animal Breeding and Healthy Husbandry, College of Animal Science and Veterinary Medicine, Tianjin Agricultural University, Tianjin 300384, China

Funding

Tianjin Natural Science Foundation, China (20JCQNJC00650), the Scientific Research Program of Tianjin Education Commission, China (2018KJ188), Lei Pu

the Key R&D and Transformation Program of Qinghai Province Science and Technology Assistance Program for Qinghai, China (2023-NK-141), Guofang Wu

the Tianjin pig industry technology system innovation team, China (ITTPRS2021006), Jianbin Zhang

the National Swine Industry Technology System (CARS-35), the Agricultural Science and Technology Innovation Program (ASTIP-IAS02), Xin Liu

the National undergraduate innovation and entrepreneurship training program, the Tianjin Key Laboratory of Green ecological feed (TJ202302), and the Tianjin excellent agricultural science and technology special commissioner and support projects. Lei Pu

Lei Pu: Conceptualization; Methodology; Validation; Formal analysis; Investigation; Funding acquisition.

Guofang Wu: Funding acquisition.

Jianbin Zhang: Writing—review and editing.

Xin Liu: Methodology; Validation; Project administration

Dear reviewer： 

Thank you very much for your timely reply and very professional comments and useful suggestions, which has significantly raised the quality of the manuscript. We have given careful consideration to every comments and suggestions. The following is a peer-to-peer response and revision to the various comments and suggestion.

Questions and suggestion 1: Why did you choose GPX2 not GPX1, GPX3 or GPX4? In other words, what is the difference between GXP1 and the other GPXs? Please clarify this in the introduction. Did you check the expression GPXs in pigs? Is GPX2 expression greater or lower than the other GPXs? Also, did you check the expression of the other GPXs when you overexpressed GPX2 in your cells?

Author’s answer 1: Thank you very much for such a professional question. Previous studies have reported that GPX2 is an important candidate gene for RFI traits. The authors also found that the type of SNP mutation in the GPX2 gene was significantly associated with backfat thickness and average daily gain in pigs (Pu et al., 2022). These contents are also added to the introduction of this article according to the experts' suggestions. GPX2 is a member of gastrointestinal GPX, but numerous studies have found that it plays an important role in muscle and adipose tissue. The pig which have the different genotypes of GPX2 have the difference expression of GPX2 in muscle and backfat tissues. It is moderately expressed in muscle and adipose tissue. The specific results are shown below (Pu et al., 2022). For this reason, this study mainly studied the function of GPX2. 

(Results from Pu et al., 2022)

Questions and suggestion 2: Since GPX2 plays critical role in regulating oxidative stress, why did you overexpress it in adipocytes and myocytes not liver? Liver is one of the major sites for ROS production. Have you checked the expression of GPX2 in different tissues in pigs? 

Author’s answer 2: Our previous results showed that GPX2 was more highly expressed in adipose tissue, muscle tissue, lungs and ileum of pigs, while there was no significant difference in the liver(Pu et al., 2022). From the aspect of animal nutrition, GPX2 has been partially reported in intestinal. But it has not been reported in adipocytes and myocytes of pig. So that's why we did this study.

Questions and suggestion 3: Why did you choose overexpress of GPX2 rather than knockout or knockdown it?

Author’s answer 3: The reviewer's question is so professional. I've also always wanted to study the effects of knockdown the GPX2 gene. Because the CDs regions of the GPX2 was very short (just over 500 bp). We also consulted a number of biological companies, but they were unable to design suitable siRNA fragments, so this part of the experiment could not be carried out.

Questions and suggestion 4: Is there enough evidence suggests that selenium supplementation can increase the expression of GPX2? If not the conclusion about "selenium as a nutritional supplement to promote human subcutaneous fat reduction and muscle growth" should be revised.

Author’s answer 4: The reviewer's question are very precise. This article does not study the functions of selenium, so the text in the article is not rigorous. This part of the content will be deleted.

Questions and suggestion 5: Make sure the format is consistent. For example, Line 88: "Porcine skeletal muscle cells culture and myogenic differentiation" is not bold.

Author’s answer 5: I am very sorry for such an inappropriate formatting error. I will correct this error and further check the article carefully to avoid relevant formatting errors.

Questions and suggestion 6: I feel something is missed between oxidative stress and lipid metabolism. In the introduction you mentioned GPX2 plays an important role in regulating oxidative stress, but how does this lead you to determine whether overexpression of GPX2 may change lipid metabolism? Please make the logic more clear and then readers can understand your story easier.

Author’s answer 6: Thanks for the reviewer's advice. This part of the content is really not clear. I will revise it in the article. The SNPs of GPX2 gene was found to be significantly associated with backfat thickness in Duroc pigs. Because GPX2 is a gene associated with oxidative stress, we studied the effects of overexpression of GPX2 on oxidative stress, lipid metabolism and glycolysis. In the supplement figure of this paper, it is listed that overexpression of GPX2 gene does not affect the expression of NRF2 protein in adipocytes and myocytes of pig. Combined with the results in the text, we concluded that GPX2 gene promotes lipid degradation by inhibiting lipid deposition in adipocytes, and at the same time leads to abnormal of glycolysis. 

The logic in the article is indeed not clear enough, and we will further modify the introduction and discussion.

Questions and suggestion 7: Based on your results, it seems that GXP2 plays different roles in different type of cells. Therefore, you should make careful conclusions that increased expression of GPX2 might be good for human health. Because overexpression of GPX2 in other cells such as hepatocyes or immune cells might not be good for human health. More research is required to further demonstrate the role of GPX2 for overall health in an animal or human.

Author’s answer 7: I strongly agree with the opinion of the reviewer. The description that GPX2 overexpression promotes human health is really inadequate. The authors agreed that it would be more appropriate for this section to be deleted. 

Questions and suggestion 8: Your results indicate that oxidative stress can increase GPX2. Besides, overexpression of GPX2 seems to be good for health since it inhibit adipocyte proliferation and promotes muscle cell growth. Therefore, do you think mild oxidative stress might be good for health?

Author’s answer 8: In fact, GPX2 expression increases after oxidative stress, when the body activates repair mechanisms. GPX2 belongs to the family of glutathione peroxidases, which have an antioxidant effect and reduce oxidative stress. Oxidative stress is known to be bad for animal growth. Therefore, we can think from another angle, through genetic methods, find the SNPs that can promote GPX2 expression, and achieve the selection of individuals with high GPX2 genes, which is the previous research results of our team(Pu et al., 2016). We will also revise this part in the discussion.

Thanks again to the reviewer for their professional suggestions and comments. We have revised the article accordingly. If the editorial department has any questions, please do not hesitate to contact us directly. 

Have a nice day!

Sincerely

Lei Pu 

E-mail: pulei87@126.com

Tianjin Key Laboratory of Agricultural Animal Breeding and Healthy Husbandry, College of Animal Science and Veterinary Medicine, Tianjin Agricultural University, Tianjin 300384, China.

Reference 

Pu L, Luo Y, Wen Z, Dai Y, Zheng C, Zhu X, et al. GPX2 Gene Affects Feed Efficiency of Pigs by Inhibiting Fat Deposition and Promoting Muscle Development. Animals (Basel). 2022;12(24). Epub 2022/12/24. doi: 10.3390/ani12243528. PubMed PMID: 36552449; PubMed Central PMCID: PMCPMC9774625.

---

## [Decision Letter · Decision Letter 1]

17 Nov 2023

PONE-D-23-26748R1Overexpression of GPX2 gene regulates the development of porcine preadipocytes and skeletal muscle cells through MAPK signaling pathwayPLOS ONE

Dear Dr. Pu,

Thank you for submitting your manuscript to PLOS ONE. After careful consideration, we feel that it has merit but does not fully meet PLOS ONE’s publication criteria as it currently stands. Therefore, we invite you to submit a revised version of the manuscript that addresses the points raised during the review process.

We look forward to receiving your revised manuscript.

Kind regards,

Juan J Loor

Academic Editor

PLOS ONE

Journal Requirements:

Reviewers' comments:

Reviewer's Responses to Questions

**Comments to the Author**

1. If the authors have adequately addressed your comments raised in a previous round of review and you feel that this manuscript is now acceptable for publication, you may indicate that here to bypass the “Comments to the Author” section, enter your conflict of interest statement in the “Confidential to Editor” section, and submit your "Accept" recommendation.

Reviewer #1: All comments have been addressed

2. Is the manuscript technically sound, and do the data support the conclusions?

Reviewer #1: Partly

3. Has the statistical analysis been performed appropriately and rigorously? 

Reviewer #1: Yes

4. Have the authors made all data underlying the findings in their manuscript fully available?

Reviewer #1: Yes

5. Is the manuscript presented in an intelligible fashion and written in standard English?

Reviewer #1: No

6. Review Comments to the Author

Reviewer #1: English needs to be polished. Please find native English speaker to revise the manuscript and make it more readable.

7. PLOS authors have the option to publish the peer review history of their article (what does this mean?). If published, this will include your full peer review and any attached files.

Reviewer #1: **Yes: **Yusheng Liang

---

## [Author Response · Author response to Decision Letter 1]

7 Dec 2023

Dear reviewer：

Thank you very much for your reply about professional comments and helpful suggestions, which have significantly raised the quality of the manuscript. 

We have studied the editor’s requirements. We have carefully modified the format errors of the references. We have also inquired the rejection and withdrawal references mentioned by the editor, and All references have no problem. The editor also suggested the paper English language needs to be polished. Need find native English speaker to revise the manuscript and make it more readable. We have asked an native English speaker who is animal science professionals to revise this article. After modification, the language of the article has been standardized a lot.

Thanks again to the reviewer for their suggestions and comments. We have revised the article accordingly. If the editorial department has any questions, please do not hesitate to contact us directly.

Have a nice day!

Sincerely

Lei Pu 

E-mail: pulei87@126.com

Tianjin Key Laboratory of Agricultural Animal Breeding and Healthy Husbandry, College of Animal Science and Veterinary Medicine, Tianjin Agricultural University, Tianjin 300384, China.

---

## [Decision Letter · Decision Letter 2]

20 Dec 2023

PONE-D-23-26748R2Overexpression of GPX2 gene regulates the development of porcine preadipocytes and skeletal muscle cells through MAPK signaling pathwayPLOS ONE

Dear Dr. Pu,

Thank you for submitting your manuscript to PLOS ONE. After careful consideration, we feel that it has merit but does not fully meet PLOS ONE’s publication criteria as it currently stands. Therefore, we invite you to submit a revised version of the manuscript that addresses the points raised during the review process.

We look forward to receiving your revised manuscript.

Kind regards,

Juan J Loor

Academic Editor

PLOS ONE

Journal Requirements:

Reviewers' comments:

Reviewer's Responses to Questions

**Comments to the Author**

1. If the authors have adequately addressed your comments raised in a previous round of review and you feel that this manuscript is now acceptable for publication, you may indicate that here to bypass the “Comments to the Author” section, enter your conflict of interest statement in the “Confidential to Editor” section, and submit your "Accept" recommendation.

Reviewer #1: All comments have been addressed

2. Is the manuscript technically sound, and do the data support the conclusions?

Reviewer #1: Partly

3. Has the statistical analysis been performed appropriately and rigorously? 

Reviewer #1: Yes

4. Have the authors made all data underlying the findings in their manuscript fully available?

Reviewer #1: Yes

5. Is the manuscript presented in an intelligible fashion and written in standard English?

Reviewer #1: Yes

6. Review Comments to the Author

Reviewer #1: 1. Western blot: I noticed that repeated showing the western blot of GPX2 overexpression. I assume that you used the same cell lysates blotted many proteins. It is fine and you don't need to showing the same GPX2 overexpressed bands all the time.

2. IF staining: Figure 6 could you please zoom in the IF staining and find more representative images to show the location of nuclei? Based on the images you provided, it is hard for the reviewer to see the difference.

3. The resolution of your figure is pretty low. Figures look blur. Please fix it.

7. PLOS authors have the option to publish the peer review history of their article (what does this mean?). If published, this will include your full peer review and any attached files.

Reviewer #1: **Yes: **Yusheng Liang

---

## [Author Response · Author response to Decision Letter 2]

2 Jan 2024

Dear reviewer：

Thank you sincerely for your response regarding the professional comments and helpful suggestions, which have significantly enhanced the manuscript's quality. We have thoroughly considered each comment and suggestion. The subsequent peer-to-peer response and revision address the comments and suggestions in detail.

Questions and suggestion 1: Western blot: I noticed that repeated showing the western blot of GPX2 overexpression. I assume that you used the same cell lysates blotted many proteins. It is fine and you don't need to showing the same GPX2 overexpressed bands all the time.

Author’s answer 1: The reviewer's question are very precise. We used the same cell lysates to blot many proteins. But to make it easier to distinguish, we replaced the same GPX2 overexpressed band.

Questions and suggestion 2: IF staining: Figure 6 could you please zoom in the IF staining and find more representative images to show the location of nuclei? Based on the images you provided, it is hard for the reviewer to see the difference.

Author’s answer 2: The reviewer's question is so professional. We analyzed 12 staining images about each of the NC group and GPX2 overexpression group, and observed overexpressing GPX2 had an increased effect on the differentiation index and fusion index in porcine skeletal muscle cells, but no significant difference. We replaced the images with clearer images, but the conclusion did not change.

Questions and suggestion 3: The resolution of your figure is pretty low. Figures look blur. Please fix it.

Author’s answer 3: We found the corresponding original figure and modified the figure format to ensure that the resolution of the figure met the requirements.

Thanks again to the reviewer for their suggestions and comments. We have revised the article accordingly. If the editorial department has any questions, please do not hesitate to contact us directly.

Have a nice day!

Sincerely

Lei Pu 

E-mail: pulei87@126.com

Tianjin Key Laboratory of Agricultural Animal Breeding and Healthy Husbandry, College of Animal Science and Veterinary Medicine, Tianjin Agricultural University, Tianjin 300384, China.

---

## [Editor Report · Decision Letter 3]

31 Jan 2024

Overexpression of GPX2 gene regulates the development of porcine preadipocytes and skeletal muscle cells through MAPK signaling pathway

PONE-D-23-26748R3

Dear Dr. Pu,

We’re pleased to inform you that your manuscript has been judged scientifically suitable for publication and will be formally accepted for publication once it meets all outstanding technical requirements.

Kind regards,

Juan J Loor

Academic Editor

PLOS ONE
---

## [Editor Report · Acceptance letter]

29 Apr 2024

PONE-D-23-26748R3 

PLOS ONE

Dear Dr. Pu, 

I'm pleased to inform you that your manuscript has been deemed suitable for publication in PLOS ONE. Congratulations! Your manuscript is now being handed over to our production team.

Kind regards, 

on behalf of

Dr. Juan J Loor 

Academic Editor

PLOS ONE